# Agri-Food Waste Recycling for Healthy Remedies: Biomedical Potential of Nutraceuticals from Unripe Tomatoes (*Solanum lycopersicum* L.)

**DOI:** 10.3390/foods13020331

**Published:** 2024-01-20

**Authors:** Vincenzo Piccolo, Arianna Pastore, Maria Maisto, Niloufar Keivani, Gian Carlo Tenore, Mariano Stornaiuolo, Vincenzo Summa

**Affiliations:** Department of Pharmacy, University of Naples Federico II, Via Domenico Montesano, 49, 80131 Naples, Italy; vincenzo.piccolo3@unina.it (V.P.); arianna.pastore@unina.it (A.P.); maria.maisto@unina.it (M.M.); niloufar.keivani@unina.it (N.K.); giancarlo.tenore@unina.it (G.C.T.); mariano.stornaiuolo@unina.it (M.S.)

**Keywords:** green tomatoes, glycoalkaloids, α-tomatine, nutraceuticals

## Abstract

Unripe tomatoes represent an agri-food waste resulting from industrial by-processing products of tomatoes, yielding products with a high content of bioactive compounds with potential nutraceutical properties. The food-matrix biological properties are attributed to the high steroidal glycoalkaloid (SGA) content. Among them, α-tomatine is the main SGA reported in unripe green tomatoes. This review provides an overview of the main chemical and pharmacological features of α-tomatine and green tomato extracts. The extraction processes and methods employed in SGA identification and the quantification are discussed. Special attention was given to the methods used in α-tomatine qualitative and quantitative analyses, including the extraction procedures and the clean-up methods applied in the analysis of *Solanum lycopersicum* L. extracts. Finally, the health-beneficial properties and the pharmacokinetics and toxicological aspects of SGAs and α-tomatine-containing extracts are considered in depth. In particular, the relevant results of the main in vivo and in vitro studies reporting the therapeutic properties and the mechanisms of action were described in detail.

## 1. Introduction

Tomatoes (*Solanum lycopersicum* L.) are among the most cultivated and consumed vegetables worldwide. The Food and Agriculture Organization Corporate Statistical Database (FAOSTAT) reported tomato global production exceeding 180 million metric tons in the two-year period of 2020–2021 [1]. Tomatoes are consumed fresh or processed in a wide variety of products including sauces, creams, concentrates, dried fruits, and pastes [2]. As a consequence of this massive demand, tomato industries produce about 15 million tons of waste pre- and post-processing [2]. The pre-processing waste, amounting about 1–6% of the total tomato production, includes damaged tomatoes and green tomatoes [3]. During red tomato harvesting, green tomatoes are separated by a color-sorting machine and discarded. The main post-processing waste is tomato pomace, a mixture of seeds, peels, fibrous parts, and a small part of the pulp [4]. This waste represents up to 2 million tons of organic material and about 5% of processed tomato weight [5]. Tomato agricultural by-products represent a disposal cost for industries and affect the environment. Therefore, governments promote the recovery of agricultural wastes, granting campaigns aimed at by-product recycling [6,7].

Green tomato waste is mainly recycled to prepare cocktail sauces, snacks, and other food products [8]. Green tomato juice can be also used to replace sugar-rich juice for the preparation of beverages or for animal feeding [9]. However, considering the nutraceutical properties of its secondary metabolites, tomato waste is being used for the isolation of natural compounds to include in supplements, additives, and pharmaceuticals. Red tomato waste has been extensively used for the extraction of lycopene [10]. Differently, green tomatoes are just starting to be considered for pharmaceutical or nutraceutical formulation, despite unripe and ripe tomatoes both displaying interesting nutraceutical components [11]. The main composition differences between the green and red maturation stages are related to secondary metabolite classes, such as steroidal glycoalkaloids (SGAs), polyphenols, carotenoids, ascorbic acid, and chlorophylls. The polyphenolic content is highly variable between green and red tomatoes. High polyphenolic contents have been reported in green tomatoes, particularly for chlorogenic acid and rutin [12]. The content of the two components decreases as the fruit ripens, but increases in ripe red tomatoes. In this stage, the main polyphenols are naringenin and homovanillic acid [12]. Carotenoids represent a class of distinctive tomato secondary metabolites that differ qualitatively and quantitatively according to the degree of fruit ripeness. The most abundant carotenoid in immature tomatoes is lutein (about 45% of the total carotenoid content), but violaxanthin, neoxanthin, and beta-carotene are also representative [13]. In ripe red tomatoes, lycopene is the most distinctive, accounts for between 59 and 91% of total carotenoid content, and gives the typical red color; beta-carotene and lutein are also relevant [14]. The high lycopene content of ripe red tomatoes, which is completely absent in unripe green tomatoes, justifies the use of waste products from ripe tomatoes as a source for the extraction of carotenoids [13]. On the contrary, green tomatoes are rich in essential nutrients such as vitamins (e.g., C, K, and B-complex vitamins), minerals (e.g., potassium, magnesium), and dietary fibers, all contributing to a balanced diet. Ascorbic acid, known as vitamin C, is a powerful antioxidant with several benefits to human health. The content of vitamin C in tomato fruits is dependent from the degree of ripeness [8]. Green tomatoes display a vitamin C content of about 25 mg/100 g of fresh fruit. This content increases as the fruit ripens, but slightly decreases in the ripe red tomato stage, reaching a content of about 10–20 mg per 100 g of fresh fruit [15]. However, data in the literature do not agree on the link between vitamin C content and the degree of ripeness [11]. Green tomatoes also contain phytochemicals like β-carotene, as well as chlorophyll, with antioxidant and anti-inflammatory properties. Therefore, they reduce the risk of dysmetabolism and cardiovascular diseases as diet supplements [16,17,18]. Chlorophylls are secondary metabolites that give the typical green color to unripe fruits. Their content decreases dramatically as the fruit ripens [8].

SGAs represent a class of metabolites specifically found in Solanaceae family plants. The most abundant SGA in immature tomatoes is α-tomatine, but dehydrotomatine, hydroxytomatine, acetoxytomatine, and esculeoside A are also representative [19]. This review summarizes data from the literature which describe the extraction methods from waste, the analysis protocols, and the main pharmacological activities of green tomato extracts titrated in α-tomatine. In addition, a comprehensive overview of the biological effects of unripe green tomato extracts containing α-tomatine, pure α-tomatine, and its aglyconic part, tomatidine, is reported, showing their potential use for human health and dietary supplements. 

## 2. Unripe Green Tomatoes: A Source of Glycoalkaloids

SGAs are a class of secondary metabolites produced by some plants in the Solanaceae family, such as tomatoes, potatoes, and eggplants. They are distributed in different botanical parts, which include the leaves, flowers, stems, roots, and unripe fruits [20]. Tomatine was firstly isolated by Fontaine et al., in tomato leaves and represents the main SGA isolated from the whole tomato plant [21]. Tomatine was originally isolated as a mixture of two SGAs, α-tomatine and dehydrotomatine, which differ based on a double bond in the aglycon moiety across positions 5 and 6. However, the chemical composition of tomatine commercial standards were characterized as a mixture of 90% α-tomatine and 10% dehydrotomatine in 1994 [22]. Therefore, all the studies conducted up to that year on tomatine actually referred to a mixture of the two compounds, presumably with the ratio 9/1. α-tomatine (C_50_H_83_NO_21_; molecular weight: 1034.2) is characterized by a nitrogen-containing spirostanic aglycone, called tomatidine (C_27_H_45_NO_2_; molecular weight: 415.7), attached to a tetrasaccharide unit in position 3β (*S* configuration). It is referred to as lycotetraose, which consists of two D-glucose and single D-galactose and D-xylose units [23]. The chemical structure is reported in Figure 1. α-tomatine belongs to the class of phytoanticipins, which are natural antimicrobial compounds that protect the plant against potential pathogens as fungi [24]. Although α-tomatine represents the most abundant tomato SGA, several secondary metabolites have been identified, both as biosynthetic precursors and as products of α-tomatine metabolism in the plant. The chemical diversity of α-tomatine derivatives is derived from some chemical modifications, such as isomerization, hydroxylation, acetylation, and glycosylation [25]. α-tomatine is present in all tomato parts, and the fruit concentration depends on the stage of maturity. Indeed, this compound is accumulated in unripe and green tomatoes, whereas during fruit maturation, it drastically decreases [26]. During the transition from green to red fruit, it is metabolized to esculeoside A, which represents the main mature tomato SGA [25]. As described in the literature, the α-tomatine concentration in tomato plants is highly variable, and different quantification values are reported. Some authors have identified the fruit as the botanical part with the highest SGA concentration [27]. Instead, other authors have reported a higher α-tomatine concentration in the stems and leaves [26]. Furthermore, the α-tomatine content is influenced by several parameters, including the tomato cultivar [28], the cultivation method (e.g., organic, conventional), and the presence of organic nutrients for plant health [29]. The α-tomatine contents reported by some authors for green tomatoes and various botanical parts of the *Solanum lycopersicum* L. plant are summarized in Table 1.

## 3. α-Tomatine Extraction from Green Tomatoes

α-tomatine extraction represents a key process for the correct titration of unripe tomato extracts. Tomato SGAs are usually extracted by grinding the samples with a mortar and pestle, or blending and then extracting analytes with polar solvent systems [19]. The protocols for unripe tomato analysis involve adding extraction mixtures to a specified amount of fresh tomatoes or lyophilized powdered tomatoes and stirring the mixture for a certain period at a controlled temperature. The most common extraction solvents are acidic aqueous or organic solutions (e.g., methanol, acetonitrile–methanol, methanol/chloroform, or tetrahydrofuran). SGAs are basic compounds due to the nitrogen atom in the spirostanic ring. Indeed, α-tomatine solubility in water depends on the pH of the solution (6 mM at pH = 5; 1 mM at pH = 6; 0.04 mM at pH = 7; and 0.03 mM at pH = 8) [44]. Acetic acid improves α-tomatine extraction by protonating the nitrogen atom and increasing the water solubility. However, acetic acid is mainly used in dried samples, while organic solvents (e.g., methanol or chloroform) are preferable with fresh tomatoes. Indeed, fresh samples contain a water content that is sufficient to solubilize the alkaloids during the extraction [45]. After the extraction, the solution can be (1) centrifuged to remove the precipitated components [22,28,29,31,32,33,37,38,46,47,48,49]; (2) concentrated under vacuum and acidified with a solution of hydrochloric acid (0.2 N) [35,50,51]; (3) or subjected to both procedures [36,52]. The samples obtained can be directly analysed [28,32,46,48] or subjected to partial purification procedures. The most common approaches to partial purification include (1) the clean-up of the samples with solid-phase extraction (SPE) [22,29,31,33,34,37,38,40,41,51,53]; (2) the precipitation and centrifugation of SGAs in basic conditions through the addition of an ammonium solution [14,35,36,50,52,54]; (3) or liquid–liquid extraction (LLE) with organic solvents [22]. The purification approach with ammonium hydroxide allows SGAs to be selectively purified. However, the precipitation is not quantitative and is characterized by a low recovery rate [45]. Instead, the SPE purification approach allows for greater reproducibility, but requires the validation of the recovery parameters for a correct quantification [51]. Various SPE cartridges are commercially available and are used in SGA analysis. The most common sorbent type is the octadecyl phase (C_18_), but other SPE sorbents include a sulfonic acid cation exchanger (SCX), a macro porous copolymer (Oasis HLB), a polar cyanopropyl (CN), aminopropyl bonded silica (NH_2_), combined packings (CN/SiOH; NH_2_/C_18_), and mixed sorbent phases as a combination of octyl and SCX (Certify) [55,56]. The SPE clean-up procedures include a first step of absorption of SGAs to the sorbent, a second step of washing the impurities off the phase, and finally, the elution of SGAs with an organic solvent, generally methanol. Impurity washing can be performed with water to remove fibers and sugars, or with aqueous organic and ammonium mixtures to elute acidic interferents as polyphenols [22,37]. The C_18_ SPE requires the loading of only aqueous extracts to allow the absorption of SGAs at the hydrophobic phase. Therefore, the C_18_ SPE is frequently suitable for acidic aqueous extracts that require a clean-up or the elimination of possible interferents in sample analysis [29,51,53]. The extracts obtained with organic solvents require the evaporation of the organic solvent and then its solubilization in water before loading on the cartridge. To ensure SGAs’ solubility and the loading of the sorbent cartridge, heptanesulfonic acid can be added as an ion-pairing reagent, improving the linkage with the C_18_ SPE phase [42]. α-tomatine recoveries with different types of SPE are greater than those of other SGAs (e.g., α-solanine, α-chaconine) and are close to 100%. However, the use of the SCX sorbent phase ensures greater selectivity for SGA purification, allowing a reduced matrix effect during the analysis. The sulfonic acid group is strongly acidic and interacts with basic species, as the nitrogen group of SGAs, improving the efficiency of the purification [33,51]. The SCX SPE requires different conditions for the interferents’ clean-up and the elution of SGA fractions compared to C_18_ SPE purification. The clean-up is carried out using aqueous organic mixtures of methanol, generally at 5–10%, which remove non-basic interferents such as polyphenols. Instead, SGA elution is performed with basic ammonia mixtures in organic solvents (e.g., 2.5–5% ammonium in methanol) [57,58].

Another approach for sample clean-up includes the liquid–liquid extraction (LLE) of the aqueous ammonia solution with a water-saturated 1-butanol solution to recover the 1-butanol layer enriched by the SGA fraction. 1-butanol LLE extraction can be used after a C_18_ SPE clean-up to obtain a robust purification of the samples from matrix interferents [37]. Ultrasound-assisted extraction (UAE) is commonly associated with the other extraction steps to improve the extraction yield. UAE is a flexible, low cost, simple, and scalable non-conventional technique. It is based on the cavitation principle, which allows cell wall disruption with the extraction of bioactive compounds. The extraction procedure can also be conducted in high throughput in order to reduce the analysis times and maximize the extraction yield. A validated method suitable for α-tomatine and tomatidine extraction has been described, requiring an approximate preparation time for each sample of 1.25 min, with a α-tomatine extraction recovery close to 100% and without clean-up procedures [19]. α-tomatine extraction protocols are applied for the extraction and quantification of other tomato SGAs [19,47]. These methods are based on the use of pH modifiers (e.g., formic acid, acetic acid, ammonium hydroxide), modifying compounds’ solubility for the ionization of the nitrogen atom, which is in common with all α-tomatine analogues. Some protocols are reported for the extraction of acetoxytomatine [19,47] and tomatidine [19,35,40,58,59]. Table 2 summarizes the conditions of the extraction methods applied over the years to extract α-tomatine.

An essential point in the analytical validation of an extraction is the recovery study of the active ingredient from the matrix. Although several papers report methods for α-tomatine extraction, few authors have adequately supported them with recovery studies. However, it seems that using acidified solvents guarantees a very high recovery and adequate analytical accuracy [28,33,42]. Therefore, the use of acidified hydroalcoholic mixtures, in combination with UAE, represents an efficient and fast method for the quantification of α-tomatine. The use of more drastic clean-up procedures significantly reduces the extraction recovery, making the sample unsuitable for quantitative analysis but only for qualitative characterization. For example, an extractive procedure with two consecutive clean-up processes has been reported in the literature, applying a purification method with a C_18_ SPE and with a 1-butanol LLE [22]. However, this procedure resulted in a very clean α-tomatine sample, unfortunately with very low recovery. Since an independent study reported a high recovery through SPE C18 purification [51], it is reasonable to conclude that the low recovery is due to the clean-up with the 1-butanol LLE.

Although α-tomatine extraction methods are reported in detail, to our best knowledge, examples of scalability are not described in the literature. Therefore, it is not possible to verify whether the previously described conditions represent an exhaustive system for glycoalkaloid extraction on a multigram scale.

Due to the structural similarity between α-tomatine and secondary SGAs, it could be hypothesized that they are effectively extracted in the same conditions as α-tomatine. However, analytical standards for most secondary tomato SGAs are not available, avoiding extraction validation and the development of the analytical methods.

## 4. α-Tomatine Analysis Methods

Over the years, several techniques have been applied for α-tomatine identification and quantification. High-performance liquid chromatography (HPLC) is the analytical approach that has been the most widely employed. One of the first detectors coupled with HPLC that was used for α-tomatine analysis was gas chromatography [60]. Due to the polarity of this compound, it could not be analysed directly in an intact form, but required hydrolysis reactions of the sugar component and functionalization to increase the volatility [60]. The analysis can be carried out either on lycotetraose sugars or on aglycones. For example, an identification protocol through SGA hydrolysis was described based on the reduction of monosaccharides to alditols, which are subsequently acetylated [38]. Although GC is a rather sensitive technique, it suffers from several disadvantages. SGAs are derivatives that often share the same aglycone and differ in terms of the tetrasaccharide moiety. Therefore, they can generate the same aglycone during the hydrolysis reaction, leading to an error in identification and quantification in real samples [11]. A second problem is the use of hydrolysis and derivatization steps, which require stringent validation to be used for quantitative purposes. SGAs with tomatidine-type aglycons generate non-specific GC fragmentation that is difficult to monitor for quantitative analysis. To solve this problem, a protocol of the trimethylsilylation and pentafluoropropionylation of tomatidine has been proposed [61]. Although this approach made the GC fragmentation spectrum clearer for identification, the reaction was not complete, and the authors reported the product of the two reactions and the product of the silylation step. The incomplete aglycone reaction may represent an issue during SGA quantification.

The most commonly used approach for the analysis of α-tomatine is liquid chromatography combined with an electron spray ionization mass spectrometer (LC/ESI-MS), a quadrupole time-of-flight mass spectrometer (Q-TOF MS), and Fourier transform ion cyclotron resonance mass spectrometry (FTICR-MS) [25,31,40,62,63]. The soft ionization of these techniques allows for the analysis of highly polar, high-molecular-weight, and thermally unstable compounds in their intact form [11]. Furthermore, SGA nitrogen atoms make the analysis extremely sensitive in positive acquisition mode, achieving a sensitivity with an α-tomatine LOQ value of 1.1 femtomoles [19]. Qualitative experiments (e.g., full MS, MS/MS, DDA) are the most widely used approach for the identification of α-tomatine and other SGAs from food matrices [31,47] and biological samples [64,65,66]. In addition, the application of more sensitive experiments for quantitative analysis (e.g., MRM) and chromatographic separation allows the sample to be injected without partial purification procedures (e.g., SPE), which are time-consuming and unsuitable for quantitative analysis [19,28,29,30,31,46,48,49]. The main disadvantages are related to the dependence on ionization, which can alter the molecule response over time, suffers from ion suppression, and requires frequent instrument calibration [33]. However, unlike GC, several published papers report the validation of α-tomatine analysis with these techniques.

Although gas chromatography (GC) and mass spectrometry (MS) are widely used approaches for α-tomatine analysis, they are not often available in laboratories. Therefore, some analytical methods have been developed with reverse-phase high-pressure liquid chromatography (RP-HPLC) using other detectors, such as pulsed amperometry (PAD), ultraviolet (UV), diode arrays (DADs), evaporative light scattering (ELS), and refractive index detectors (RIDs). The pulsed amperometric detection (PAD) represents one of the first techniques used for α-tomatine analysis. This detector was used for the first quantitative analysis of α-tomatine in tomatoes [22], the different parts of the plant [38], and processed tomato products [37]. Although PADs are a more sensitive analytical technique than other detectors (e.g., UV and DADs), HPLC analysis can generate a non-linear response due to the overloaded detector cell at high concentrations [38]. The ultraviolet (UV) and diode array (DAD) detectors are the most widely used techniques due to their easy use and low cost of analysis [33,34,35,36,51,53,58,67]. However, α-tomatine is detected around 200 nm due to the lack of UV-visible functional groups, a wavelength with low specificity. Therefore, HPLC analysis requires the use of clean-up techniques (e.g., SPE) to reduce the number of possible interferents and is characterized by low sensitivity. For example, an LOD value of 2.5 µg for α-tomatine analysis through HPLC-UV has been reported in the literature [38]. In addition, the analysis can only be performed using mobile phases with low UV cut-offs (e.g., acetonitrile and phosphate buffer) [68]. Given the low sensitivity of UV and DADs in the analysis of α-tomatine, some authors have found greater advantages in using these detectors for the preparative separation of SGAs from natural matrices and the subsequent analysis of the fractions with other detectors [38]. The evaporative light-scattering detector (ELSD) is easy to use, but it does not generate a linear response [60], so it has been used mainly for the identification and purity analysis of α-tomatine and other SGAs [32,48,67]. The refractive index detector (RID) is an easy-to-use detector that responds independently from parameters such as UV-visibility, volatility, or the presence of ionizable functional groups. However, RIDs lack sensitivity, cannot be used for elution gradients, and are sensitive to environmental variations in flow, pressure, and temperature [60]. Therefore, the use of RIDs is limited to the preparative separation of SGAs from natural matrices, due to the high α-tomatine content in unripe tomato extracts [69].

Although RP-HPLC methods represent the most common approach for the analysis of α-tomatine and other SGAs, several alternative methods have been developed over the years for the purification, identification, and quantification of these molecules. Thin-layer chromatography (TLC) is a technique of direct-phase chromatography that separates SGAs based on the different polarities of sugar moieties. It is used to estimate the number of molecules present in a mixture and to monitor fractions during chromatographic separation. Therefore, this technique exhibits only a qualitative value and cannot be used for quantitative purposes. However, examples of preparative TLC or high-pressure thin-layer chromatography (HPTLC) for the isolation of α-tomatine from plant matrices are reported in the literature [69,70]. Another approach used for SGA separation was capillary electrophoresis (CE), a low-cost and -speed technique which separates compounds based on their ion mobilities. CE has often been used for the separation of ionizable basic compounds such as alkaloids [71]. For example, a non-aqueous CE method coupled with an electrospray ionization mass spectrometer for α-tomatine separation and identification has been described [72]. The use of a non-aqueous solvent (e.g., methanol) increases SGAs’ solubility and avoids the use of salt buffers that are often not compatible with mass spectrometer detectors [71]. However, the CE approach has been applied only for the analysis of pure molecules of α-tomatine and tomatidine [72]. Unfortunately, to our best knowledge, papers about CE separation using real *Solanum lycopersicum* L. samples are not reported in the literature. An innovative approach would be based on the application of attenuated total reflectance infrared spectroscopy and thermogravimetric analysis to predict the α-tomatine content in tomato samples. The effectiveness of these approaches was confirmed using statistical techniques, such as partial least square regression and multiple linear regression [30]. 

## 5. Nutraceutical Potential of Green Tomatoes

The positive healthy effects of red tomatoes have been scientifically supported since 1950. Their nutritional components, vitamins (e.g., vitamin C, folic acid), electrolytes (e.g., sodium, potassium), and carotenoids (e.g., α-carotene, β-carotene, lycopene, and lutein) are renowned bioactive molecules. Lycopene is characterized by an antioxidant activity and ameliorates cardiovascular symptoms, age-related macular degeneration, and cataracts in humans [73,74,75]. 

Although last two decades have witnessed increasing scientific evidence supporting the beneficial health effects of green tomatoes and SGAs such as α-tomatine, tomatidine, and tomatidinol [76], green tomatoes have been shown to possess a wide variety of biological activities and exert antiviral, fungicide, antibiotic, anti-inflammatory, anticarcinogenic, and anti-aging effects, as summarized in Table 3 and in Figure 2 [39,52,77,78,79,80].

### 5.1. Antiviral, Antifungal, and Antibiotic Activity

SGAs are produced by plants as a defense against bacteria, fungi, viruses, and insects [102]. It is thus not surprising that the healthy properties of green tomatoes’ alkaloids are a consequence of the antibiotic power of these secondary metabolites. Leaves and immature green fruit extracts of Californian *Solanum lycopersicum* L. display antimicrobial activity against several bacteria (*Salmonella enterica*, *Staphylococcus aureus,* and *Escherichia coli* K12) [81]. Interestingly, the extract does not affect the growth of the beneficial bacteria *Lactobacillus acidophilus*, *Lactobacillus rhamnosus*, and *Lactobacillus reuteri*, which are part of the human gut microbiota [81]. A-tomatine affects the membrane permeability of many crop-infesting fungi by sequestering ergosterols, one of the main components of fungal membranes [103]. Ergosterol sequestration disrupts the membrane bilayer, ultimately causing the leakage of cell components, osmotic stress, and cell death [44,104]. Among SGAs, α-tomatine has the highest bactericidal activity against bacteria and fungi [103]. α-tomatine, isolated from young leaves of *Lycopersicon Pimpinellifolium*, showed activity against the pathogen *Fusarium caereleum* (IC_50_ = 460 µM). Further, α-tomatine included in bacterial Petri dishes completely inhibits the growth of fungal species, such as *Candida albicans* (α-tomatine-enriched extracts of green tomatoes, leaves, and stems) [81]; *Fusarium oxisporum* (IC_50_ = 40 μM); and *Cladosporium fulvum,* as well as the spore germination *of Paecilomyces Fumosoreus* (IC_50_ = 500 μM), and partially reduced of 45% the spore germination of *Beauveria brassiana* (IC_50_ = 1 mM) [82,83]. In the fungal pathogen *Fusarium*, the damage caused by this compound increases reactive oxygen species (ROS) production and leads to fungal programmed cell death [105]. α-tomatine inhibits the growth of the two parasitic protozoans *Trichomonas Vaginalis* (specific for humans) and *Trichomonas Foetus* (specific for animal farms), with IC_50_ values in the micromolar range of 8 μM and 2 μM, representing a valid therapeutic alternative to metronidazole [84]. α-tomatine is also effective against Chagas Disease caused by *Trypansoma cruzi* parasites [85], and has been shown to possess anti-ciliate activity (IC_50_ = 10.14 μM), and potent effects against *Philasterides dicentrarchi*, a scuticociliate colonizing farmed fishes [106].

### 5.2. Anti-Inflammatory Effects

Several articles have reported the anti-inflammatory effects of pure SGAs and green tomatoes’ extracts. Extracts obtained from the locular gel and serum of *Solanum lycopersicum* L. var. “Camone” (respectively, containing 61.7 ± 0.9 mg of α-tomatine/kg of FW of locular gel and 12.5 ± 0.5 mg of α-tomatine/kg of FW of serum) significantly reduce inflammation in humans, decreasing the blood inflammatory cytokine count, systolic pressure, heart rate, and aorta thickness [46]. The supplementation of 1–2% dietary tomato powder containing α-tomatine ameliorates hemato-immunological and antioxidant clinical parameters in rabbits [86]. 

Pure α-tomatine inhibits the production of the proinflammatory cytokines IL-1β, IL-6, and TNF-α in LPS-stimulated macrophages by preventing IκB degradation and ERK phosphorylation [107]. In agreement with these reports, α-tomatine has been shown to inhibit the expression of Cox-2 and iNOS and decrease the production of prostaglandin E2 (PGE2) in murine LPS-stimulated macrophages. Furthermore, α-tomatine exerts a powerful antihistaminic effect [108]. In particular, the administration of 1–10 mg of α-tomatine/kg (i.v.), 15–30 mg of α-tomatine/kg (p.o.), and 5–10 mg of α-tomatine/kg (s.c.) significantly reduces paw-induced edema in rats [89]. Its aglycone, tomatidine (10–40 µM), suppresses NF-κB activation in a dose-dependent manner by inhibiting IkB-α degradation in LPS-stimulated RAW 264.7 macrophages [88]. 

Tomatidine has also been shown to exert anti-inflammatory activity in lung tissues where, by modulating several cytokine expressions, including IL-β, IL-6, TNF-α CCL-5, MCP-1, and ICAM1 in bronchoalveolar lavage fluid, it attenuates neutrophil infiltration, increases superoxide dismutase activity and glutathione levels, and reduces myeloperoxidase expression [99]. Finally, in a rat model of osteoarthritis, tomatidine inhibited the expression of IL-1β-induced metalloproteinases, as well as the degradation of aggrecan and collagen-II, ameliorating the inflammatory condition of the animals [87].

### 5.3. Anti-Aging Effects

Green tomatoes and SGAs have shown promising anti-aging effects in many tissues, including the bones, brain, and muscles. A diet supplementation with a green tomato extract from “*Korean chal tomato*” (containing tomatidine in the amount of 1.06 ± 0.11 mg of tomatidine/100 g of dry weight) improved bone mineral density and overall bone quality in ovariectomizes rats, a model of postmenopausal osteoporosis [58]. The aglycone tomatidine inhibits osteoclastogenesis and reduces estrogenic deficiency-induced bone mass loss [90] through a mechanism that has not been fully elucidated, but that probably involves the modulation of the p53 and MAPK signaling pathways [109].

α-tomatine exerts anti-acetylcholinesterase and anti-butyrylcholinesterase activities, making SGAs eligible for the treatment of neurodegenerative diseases, such as Alzheimer’s disease. Dried extracts from the leaves of two varieties of *Solanum lycopersicum* L., “Cherry” and “Bull’s heart” (containing 640.0 ± 2.0 μg of tomatine/mg of DW extract and 9.34 ± 0.10 μg of tomatidine/mg of DW extract, and 402.0 ± 1.2 μg of tomatine/mg of DW extract and 5.39 ± 0.10 of μg tomatidine/mg of DW extract, respectively), have shown neuroprotective effects against glutamate-induced toxicity in SH-SY5Y neuroblastoma cells [34]. Tomatidine efficiently improves muscle physiology by direct binding and activating mTORC1 (mechanistic target of the rapamycin complex). It reduced atrophy in a murine model of age-related muscle aging and increases strength and exercise ability in adult mice (administration at a dosage of 25 mg of tomatidine/kg (i.p.)), promoting skeletal muscle hypertrophy and stimulating anabolism [91]. Tomatidine has been found to reduce the expression of activating transcription factor 4 (ATF4), one of the regulators of age-related muscle weakness and atrophy [110,111].

### 5.4. Anti-Tumoral Effects

Red tomato extracts from the fruits of variants of *Solanum lycopersicum* L. (var. *Sancheri premium*, *Yoyo*, *Chobok Power,* and *Rokusanmaru*) have only subtle growth inhibitory effects in several in vitro cell culture models (breast cancer (MCF-7), colon cancer (HT-29), gastric cancer (AGS), hepatocarcinoma (HepG2), and liver cancer (Chag)). However, the corresponding extracts from unripe green fruit (α-tomatine content ranging from 5.75 ± 0.29 mg of α-tomatine/100 g of FW to 31.40 ± 1.97 mg of α-tomatine/100 g of FW) inhibit the growth of several human cancer lines, such as the MCF-7, HT-29, AGS, HepG2, and Chag lines [52,78]. 

α-tomatine has been shown to exert anticarcinogenic effects in many tumor models. In vivo, it inhibits oncogenic transcriptional factors, including NF-kB family members [78,97,112,113] and its cofactor AP-1 [94], which are involved in fundamental process such as cell differentiation, cell cycle progression, and apoptosis [114]. α-tomatine administration reduced the onset of dibenzopyrene-induced liver and stomach tumors in mice [101].

In vitro, α-tomatine inhibits the growth of different cancer cell lines. In human hepatocarcinoma cultures (HepG2), α-tomatine (1 µM) promotes growth arrest with a potency similar to doxorubicin and camptothecin [92]. The mechanism of its anti-tumoral activity seems to include the disruption of tumor cell membranes, cell cycle arrest, and the inhibition of DNA replication [115]. In human hepatocarcinoma cells (HepG2), α-tomatine (30 µM) triggers apoptosis via the stimulation of caspases 3 and 7, two potent pro-apoptotic factors, as well as the inhibition of the expression of the anti-apoptotic protein Bcl-2 [93,116]. 

Interestingly, α-tomatine reduced the oncogenic properties of highly metastatic A549 lung cancer cells (1 µM) [94] through the inactivation of phosphoinositide 3-kinase/protein kinase B (PI3K/Akt) and the extracellular signal-regulated kinase 1 and 2 (ERK1/2) signaling pathways, both master regulators of cell metabolism and key players in cancer progression. α-tomatine promotes cell survival through the activation of the mTORC-1 complex, inhibiting apoptosis and stimulating protein synthesis, and modulates p53 degradation, leading to cell cycle progression [117]. Furthermore, it modulates the extracellular matrix enzymes MMP-2 and MMP-9 and the urokinase-type plasminogen activator (u-PA) enzymes, all parts of the extracellular matrix remodeling program performed by cancer cells during tumor extravasation and metastasis. The inhibition of MMP-7, MMP-2, and MMP-9 by α-tomatine has been confirmed in a human breast cancer cell line (MCF-7) (7.07 µM) [95] and in human non-small lung cancer cells (NCI-H460) (2 µM) [96]. α-tomatine (2 µM) synergizes with the chemotherapeutic agent paclitaxel, showing an antiproliferative effect on human prostate cancer (PC3) development and progression [97]. Besides the antimetastatic properties against lung cancer, a mixture of α-tomatine and dehydrotomatine (87.1 ± 1.6% and 13.0 ± 0.8%, respectively) exerts antimetastatic effects on melanoma (1 µM). Specifically, the combination of two SGAs inhibits the invasion and viability of BRAF- and V600BRAF-positive metastatic melanoma cells. In this context, α-tomatine has been shown to induce autophagy, increasing the LC3II/LC3I ratio (a common sign of autophagy) and endoplasmic reticulum stress in metastatic melanoma [98]. It also inhibits cell growth and induces apoptosis in different leukemic cell lines (5 µM), including HL-60 and K562 cells, with a potency similar to the chemotherapy agent cytosine arabinoside [78,100]. Mechanistically, it has been reported that α-tomatine downregulates survivin expression and activates the apoptotic factors Bak and Mcl-1, thus inducing the loss of potential of the mitochondrial membrane and the release of apoptosis-inducing factor (AIF). α-tomatine-dependent mechanisms for modulating apoptosis could be the result of cell membrane disruption due to the interaction between α-tomatine and cholesterol [78]. Although pure α-tomatine has proven its anti-tumoral effect in many studies, dehydrotomatine is reported to have no effect in the reduction of cell proliferation [52]. 

### 5.5. Pharmacokinetics and Toxicological Aspects of Glycoalkaloids and Green Tomato Extracts

The correlation between in vitro pharmacological activities and possible beneficial effects on human health is strictly correlated to the study of α-tomatine pharmacokinetics (e.g., absorption, distribution, biotransformation, and excretion). Although α-tomatine toxicology and the in vivo fate of other SGAs (e.g., α-solanine, α-chaconine) have been extensively studied [118], there are few data about the pharmacokinetics of tomato SGAs. For many years, α-tomatine was considered a molecule with low bioavailability. It is stable at 37 °C under acidic conditions that mimic the pH of the stomach. Furthermore, α-tomatine and cholesterol form insoluble complexes that are eliminated through feces [102]. The first attempt to study α-tomatine metabolism in an animal model was reported in 2017, identifying some metabolites in mouse plasma samples following the administration of a diet containing α-tomatine, with 10% (*w*/*w*) tomato powder [64]. In this study, α-tomatine was absent in mouse plasma, while some aglycones (e.g., tomatidine, hydroxytomatidine, didehydrotomatidine) have been detected. Therefore, a putatively metabolic mechanism of α-tomatine is mediated by the removal of tetrasaccharide moieties and phase I metabolism reactions. In another two studies, α-tomatine metabolites were reported in the skin [65] and liver of mice treated with tomato-enriched diets [66]. An independent study expanded the list of identified α-tomatine metabolites to 19 compounds, and demonstrated the presence of some sulfated and glucuronidated compounds obtained through phase II metabolism reactions. Although these studies increased the knowledge about α-tomatine metabolism, they did not quantify α-tomatine and tomatidine in animal plasma and did not identify the enzymes involved in SGA metabolism. A preliminary indication of the role of intestinal microbiota in pure-molecule α-tomatine metabolism has been described using a pig cecum model [119]. This system represents a widely used translational model due to the similarity between pigs’ and humans’ gastrointestinal microbiota. Therefore, this model is often used in ex vivo experiments to assess the effects of microbiota on the metabolism of pure molecules or extracts [120]. The incubation of α-tomatine in the active cecum suspension assessed the cleavage of tetrasaccharide moieties through the removal of single sugar units and the detection of intermediate cleaved alkaloids (e.g., β-tomatine, γ-tomatine, tomatidine). However, the degradation of α-tomatine was slow and constant in the time up to 60% of the initial concentration after 24 h [119]. 

As is well known from the literature, phase I and II metabolism reactions generally increase a drug polarity’s to accelerate its elimination rate in urine. Therefore, it is reasonable to hypothesize that sulfated and glucuronidated metabolites are involved in α-tomatine excretion. Some studies on other SGAs have detected some sulfated metabolites in human urine. One example is represented by esculeogenin B sulfonate, which derived from the metabolism of esculeogenin B that is present in ripe tomato products such as juice [121]. Although these studies partially described the in vivo α-tomatine metabolism, unfortunately, to our knowledge, there are no complete studies focused on the absorption, distribution, and excretion of this molecule. Moreover, these studies were performed on mature red tomatoes products with a low α-tomatine content, while no published data have been reported to assess pharmacokinetics using unripe tomatoes with a high α-tomatine content.

Despite clear data being missing on SGA pharmacokinetics, tomatoes and tomato extracts, including varieties highly enriched in α-tomatine, are consumed worldwide without any toxic effects [102]. Peruvian populations traditionally consume *Solanum lycopersicum* L. tomato variants (var. cerasiforme), highly enriched in α-tomatine (0.5–5 mg/g of dry weight), and there have not been reports of toxicity [43].

The excessive consumption of SGAs does, however, have harmful effects on the human body and leads to a plethora of side effects, including hypotension, gastrointestinal and neurological disorders, and coma. Among the alkaloids, α-tomatine possesses a moderate toxicity. Harmful effects can be observed with doses of 2–5 mg of α-tomatine/kg/die [44], a dosage that could be achieved either by using pure SGAs or as consequence of a massive daily consumption of green tomatoes [122]. In mice, through oral and intravenous administration, the LD_50_ of pure α-tomatine is 500 mg/kg of b.w. (body weight) and 18 mg/kg of b.w., respectively. At these dosages, α-tomatine can cause hypotension, as well as changes in the respiratory rate and red blood cell number. Through subcutaneous administration in mice, the LD_50_ of α-tomatine is superior to 1000 mg/kg of b.w. [102]. Instead, upon intraperitoneal administration, the LD_50_ of α-tomatine is 25–33 mg/kg of b.w., a dosage able to decrease diuresis in rats, in association with a decrease in the Na^+^/K^+^ serum ratio and an increase in neutrophils and corticosteroids [102]. Topically, α-tomatine seems to cause no toxicity. An ointment containing 5% α-tomatine, applied onto the skin of rabbits and rats, does not cause erythema or changes in hemoglobin and erythrocyte counts. However, when topically applied to the eye, this ointment caused conjunctivitis [123]. 

## 6. Conclusions

Waste products of the tomato industry represent a rich, natural source of α-tomatine, and the recycling of this waste represents an appealing research field to develop innovative nutraceutical products. It is thus not surprising that over-the-counter products containing green tomato extracts are starting to become popular. Among their secondary metabolites, α-tomatine exhibits significant biological activities on human health. In this review, the health-beneficial properties of pure tomato compounds (e.g., α-tomatine and tomatidine) and *Solanum lycopersicum* L. extracts in several diseases have been discussed. Besides its antioxidant power, α-tomatine-containing extracts show interesting antimicrobial, anti-inflammatory, anti-aging, and anti-tumoral activities. In vitro, the cellular and molecular mechanisms involved in green tomato pharmacological activities have been identified and proven to involve the modulation of several metabolic patterns. However, for a proper translation, these in vitro biological data require further evidence in appropriate animal models and, most importantly, in clinical trials.

Meanwhile, the accurate correlation between the beneficial effects and concentration of active ingredients through a harmonized and standardized quantification approach is a requirement. Therefore, additional properties (e.g., nutraceutical and pharmacokinetic in vitro effects) of tomato SGAs and *Solanum lycopersicum* L. extracts will require further development to investigate their effects on humans in proper in vivo studies.

## Figures and Tables

**Figure 1 foods-13-00331-f001:**
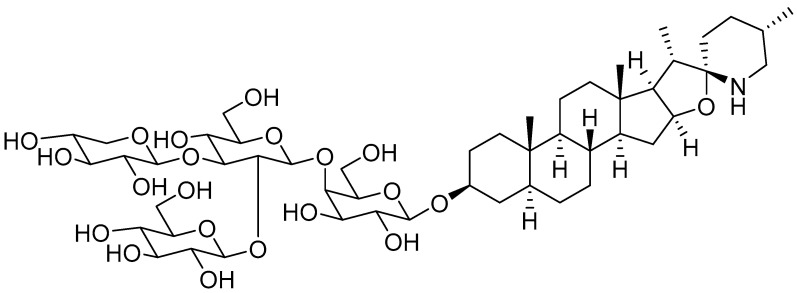
Chemical structure of α-tomatine. Structure was drawn using the chemistry software ChemDraw Professional 15.0.

**Figure 2 foods-13-00331-f002:**
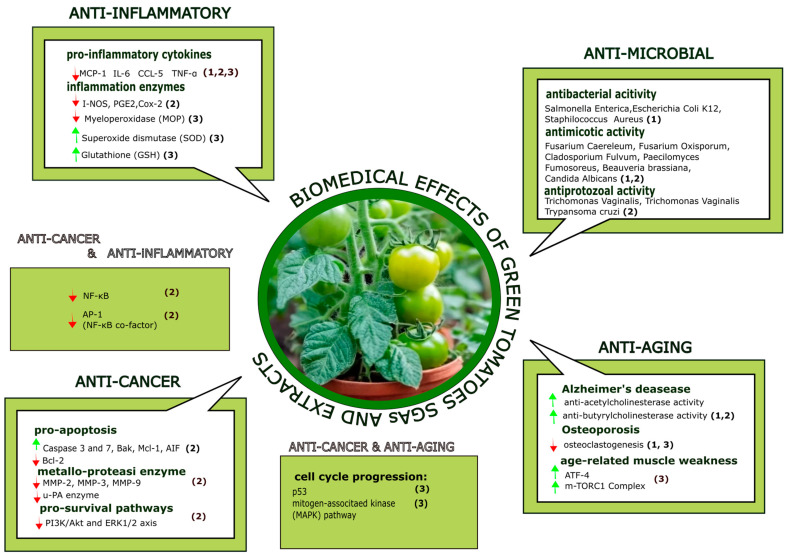
Scheme of biomedical effects of green tomato SGAs and extracts. Numbers in brackets indicate, respectively, (1) effects of green tomato extracts; (2) effects of pure α-tomatine; and (3) effects of pure tomatidine.

**Table 1 foods-13-00331-t001:** α-tomatine content reported in *Solanum lycopersicum* L. botanical parts.

Sample	α-Tomatine Content (mg/g)	Weight	Reference
Lyophilized green tomatoes	0.734 ± 0.028–0.175 ± 0.011	DW	[28]
Air-dried red tomatoes	0.112–0.005	DW	[29]
Lyophilized green tomatoes	1.993 ± 0.033–0.181 ± 0.012	DW	[30]
Lyophilized green tomatoes	34.354 ± 1.094–0.282 ± 0.019	DW	[31]
Fresh green tomatoes	1.894 ± 0.008–0.278 ± 0.004	FW	[32]
Lyophilized green tomatoes	2.240 ± 0.010–1.130 ± 0.035	DW	[22]
Lyophilized leaves	9.620 ± 0.201–6.030 ± 0.303	DW	[33]
Lyophilized leaves	0.640 ± 0.002–0.402 ± 0.001	DW	[34]
Hot-air-dried sample	2.430 ± 0.467–0.170 ± 0.002	DW	[35]
Fresh sample	0.087 ± 0.027–0.036 ± 0.014	FW	[36]
Fresh sample	0.089 ± 0.004–0.010 ± 0.001	FW	[14]
Lyophilized samples	7.435 ± 0.015–0.003 ± 0.001	DW	[37]
Fresh samples	4.985 ± 0.115–0.144 ± 0.005	FW	[38]
Fresh samples	3.610 ± 0.255–0.138 ± 0.010	FW	[39]
Fresh samples	0.308–0.003	FW	[40]
Fresh samples	0.041–0.008	FW	[41]
Fresh samples	0.065–0.006	FW	[42]
Hot-air-dried sample	31.000–0.001	DW	[43]

Abbreviations: DW: dry weight; FW: fresh weight.

**Table 2 foods-13-00331-t002:** Summary of extraction conditions reported for α-tomatine extraction.

Sample (Amount in g)	Extraction Solvent	Extraction Condition	Clean-Up	Reference
Lyophilized sample (0.1)	10 mL of acetic acid 1% in aqueous EtOH 80% (*v*/*v*)	UAE (10 min), centrifugation (5 min, 4000 rpm) for two cycles		[28,30]
Lyophilized sample (0.1)	10 mL of acetic acid 1% in aqueous EtOH 70% (*v*/*v*)	UAE (15 min), centrifugation (5 min, 4000 rpm) for two cycles		[46]
Fresh samples (10)	10 mL of methanol	UAE (1 h), centrifugation (10 min, 12,000 rpm)		[32]
Hot-air-dried sample (35)	Acetic acid 5% in EtOH	UAE (20 min)	(NH_4_)OH 5% (*v*/*v*) solution	[35]
Fresh sample	(1) H_2_O (2) Chloroform/MeOH, 2:1 (*v/v*)	UAE, centrifugation (10 min, 9800 g)	(NH_4_)OH 2% (*v*/*v*) solution	[36]
Fresh sample (1–20)	Chloroform/MeOH, 2:1 (*v*/*v*)		(NH_4_)OH 2% (*v*/*v*) solution	[14,50]
Lyophilized sample (0.5)	100 mL of aqueous acetic acid 5%	UAE (10 min)	Different SPE sorbents (C_18_, CN, SCX, Oasis HLB)	[51]
Fresh sample (2)	10 mL of formic acid 1% in aqueous MeOH 70% (*v*/*v*)	Centrifugation (10 min, 2600 g)		[48]
Lyophilized sample (0.16)	2 mL of aqueous acetic acid 5%	UAE (60 min), centrifugation (5 min, 12,000 rpm)	SPE (SCX)	[33,34,53,58]
Lyophilized sample (0.2)	4 mL of aqueous acetic acid 5%	UAE (30 min), centrifugation (5 min, 12,000 rpm)	SPE (SCX)	[57]
Air-dried sample (0.25)	10 mL of aqueous acetic acid 1%	Centrifugation (15 min)	SPE (C_18_)	[29]
Lyophilized sample (1)	30 mL of aqueous acetic acid 1%	Centrifugation (10 min, 13,300 rpm) for two cycles	SPE (C_18_), LLE with 1-butanol	[22,37]
Lyophilized sample (1)	30 mL of aqueous acetic acid 1%	Centrifugation (10 min, 13,300 rpm) for two cycles	SPE (C_18_)	[38]
Fresh sample (1–22)	100 mL of acetic acid 2% in MeOH	Centrifugation (10 min, 18,000 g)	(NH_4_)OH 2% (*v/v*) solution	[39]
Fresh sample (3–41)	100 mL of acetic acid 2% in MeOH	Centrifugation (5 min, 18,000 g)	(NH_4_)OH 2% (*v/v*) solution	[52]
Fresh sample	45 mL of MeOH	Centrifugation (5 min, 3000 rpm) for three cycles	SPE (C_18_)	[40]
Lyophilized sample (1)	20 mL of aqueous acetic acid 1%		SPE (C_18_)	[41]
Lyophilized sample (0.5)	5 mL of acetic acid 2% in MeOH	Centrifugation (15 min, 8000 rpm) for two cycles	(NH_4_)OH 25% (*v/v*) solution	[53]
Lyophilized sample (1.5)	15 mL of formic acid 1% in aqueous MeOH 80% (*v/v*)	UAE (10 min), centrifugation (10 min, 9000 rpm) for two cycles	SPE (C_18_)	[31]
Fresh sample (55)	50 mL of THF/H_2_O/Acn/Acetic acid (50/30/20/1 *v/v*)		SPE (C_18_)	[42]
Lyophilized sample (0.05)	15 mL of MeOH	Centrifugation (5 min, 3000 g)		[19]
Fresh sample (0.2)	20 mL of formic acid 0.1% in aqueous MeOH 80% (*v/v*)	Centrifugation (10 min, 20,000 g)		[47]

Abbreviations: H_2_O: water; MeOH: methanol; Acn: acetonitrile; EtOH: ethanol; THF: tetrahydrofuran; (NH_4_)OH: ammonium hydroxide; UAE: ultrasound-assisted extraction; SPE: solid-phase extraction; LLE: liquid–liquid extraction.

**Table 3 foods-13-00331-t003:** Summary of pharmacological activity of α-tomatine, tomatidine, and *Solanum lycopersicum* L. botanical part extracts.

Pharmacological Effects	Experimental Model	Extracts/Pure Molecule	α-Tomatine/Tomatidine Content	IC_50_	Reference
Antimicrobial	*Escherichia coli* K12	Dried green tomato peel extract	12 mg of α-tomatine/kg of DW; 2 mg of dehydrotomatine/kg of DW	8 mm of zone of inhibition (treatment at 10% *w/v)*	[81]
	*Salmonella enterica*	Dried green tomato peel extract	12 mg of α-tomatine/kg of DW; 2 mg of dehydrotomatine/kg of DW	7 mm of zone of inhibition (treatment at 10% *w/v)*	[81]
	*Candida albicans*	Green tomato extract Dried leaf extract Dried stem extract	n.r. 11 mg of α-tomatine/kg of DW; 7 mg of dehydrotomatine/kg of DW	11.5 mm of zone of inhibition (treatment at 10% *w/v)*	[81]
	*Salmonella enterica*	Dried leaf extract	11 mg of α-tomatine/kg of DW; 7 mg of dehydrotomatine/kg of DW	8 mm of zone of inhibition (treatment at 10% *w/v)*	[81]
	*Bacillus cereus*	Dried leaf extract	11 mg of α-tomatine/kg of DW; 7 mg of dehydrotomatine/kg of DW	13 mm of zone of inhibition (treatment at 10% *w/v)*	[81]
	*Beauveria brassiana*	α-tomatine		1 mM	[82]
	*Fusarium caereleum*	Tomatine		7–460 µM	[83]
	*Paecilomyces fumosoreus*	α-tomatine		500 µM	[82]
	*Trichomonas vaginalis*	α-tomatine		8 µM range	[84]
	*Trichomonas foetus*	α-tomatine		2 µM range	[84]
	*Trypanosoma cruzi*	α-tomatine		10.14 µM	[85]
Anti-inflammatory	In vivo rat model	Locular gel and serum extracts of *Solanum lycopersicum* L. var “Camone”	61.7 ± 0.9 mg of α-tomatine/kg of FW locular gel; 12.5 ± 0.5 mg of α-tomatine/kg of FW serum	12.40 g DW/kg rat p.o.	[46]
	In vivo rabbit model	Dried tomatoes powder	Diet of 1% and 2% tomato powder		[86]
	In vivo rat model	Tomatidine		2.5–10 μM	[87]
	In vitro murine macrophage cultures	Tomatidine		10–40 µM	[88]
	In vivo rat model	α-tomatine		1–10 mg/kg i.m. 15–30 mg/kg p.o. 5–10 mg/kg s.c.	[89]
Anti-aging	In vivo rat model	Dried green tomato extract	1.06 ± 0.11 mg of tomatidine/100 g of DW matrix		[58]
	In vitro primary cultures from rat model	Tomatidine		8 µM	[90]
	In vitro cultures of neuronal cells	Leaf extract of *Solanum lycopersicum* L. var “Cherry”	640.0 ± 2.0 µg of tomatine/mg of DW; 9.34 ± 0.10 µg of tomatidine/mg of DW	197.50 µg/mL	[34]
	In vitro cultures of neuronal cells	Leaf extract of *Solanum lycopersicum* L. var “Bull’s heart”	402.0 ± 1.2 µg of tomatine/mg of DW; 5.39 ± 0.10 µg of tomatidine/mg of DW	197.50 µg/mL	[34]
	In vivo mouse model	Tomatidine		25 mg/kg (i.p.)	[91]
Anti-tumoral	In vitro breast cancer cells (MCF-7)	Dried green tomato extract of *Solanum lycopersicon* L. var *Sancheri premium* (*S.p*), *Chobok Power* (*C.p*), *Yoyo* (*Y*), *Rokusanmaru* (*R*)	*S.p:* 10.8 ± 0.69 µg of α-tomatine mg/100 g of FW; *S.p:* 5.75 ± 0.29 µg of α-tomatine mg/100 g of FW; *Y: 8*.30 ± 0.07 µg of α-tomatine mg/100 g of FW; *R:* 11.53 ± 1.11 µg of α-tomatine mg/100 g of FW; *R:* 9.36 ± 0.32 µg of α-tomatine mg/100 g of FW	0.33 ppm; 2.41 ppm; ≤0.1 ppm; 18 ppm; 9 ppm	[52]
	In vitro colon cancer cells (HT-29)	Dried green tomato extract of *Solanum lycopersicon* L. var *Sancheri premium* (*S.p*), *Chobok Power* (*C.p*), *Yoyo* (*Y*), *Rokusanmaru* (*R*)	*S.p:* 10.8 ± 0.69 µg of α-tomatine mg/100 g of FW; *S.p:* 5.75 ± 0.29 µg of α-tomatine mg/100 g of FW; *Y: 8*.30 ± 0.07 µg of α-tomatine mg/100 g of FW; *R:* 11.53 ± 1.11 µg of α-tomatine mg/100 g of FW; *R:* 9.36 ± 0.32 µg of α-tomatine mg/100 g of FW	≤0.1 ppm; ≤0.1 ppm; 5.4 ppm; 1.3 ppm	[52]
	In vitro hepatocarcinoma cells (HepG2)	Dried green tomato extract of *Solanum lycopersicon* L. var *Sancheri premium* (*S.p*), *Chobok Power* (*C.p*), *Yoyo* (*Y*), *Rokusanmaru* (*R*)	*S.p:* 31.4 ± 1.97 µg of α-tomatine mg/100 g of FW; *S.p:* 10.8 ± 0.69 µg of α-tomatine mg/100 g of FW; *S.p:* 5.75 ± 0.29 µg of α-tomatine mg/100 g of FW; *Y: 8*.30 ± 0.07 µg of α-tomatine mg/100 g of FW; *R:* 11.53 ± 1.11 µg of α-tomatine mg/100 g of FW; *R:* 9.36 ± 0.32 µg of α-tomatine mg/100 g of FW	12.3 ppm; 3.2 ppm; 1 ppm; ≤0.8 ppm; 0.2 ppm; 0.9 ppm	[52]
	In vitro stomach cancer cells (AGS)	Dried green tomato extract of *Solanum lycopersicon* L. var *Sancheri premium* (*S.p*), *Chobok Power* (*C.p*), *Yoyo* (*Y*), *Rokusanmaru* (*R*)	*S.p:* 31.4 ± 1.97 µg of α-tomatine mg/100 g of FW; *S.p:* 10.8 ± 0.69 µg of α-tomatine mg/100 g of FW; *S.p:* 5.75 ± 0.29 µg of α-tomatine mg/100 g of FW; *Y: 8*.30 ± 0.07 µg of α-tomatine mg/100 g of FW; *R:* 11.53 ± 1.11 µg of α-tomatine mg/100 g of FW; *R:* 9.36 ± 0.32 µg of α-tomatine mg/100 g of FW	11.4 ppm; 2 ppm; 1.4 ppm; 1.7 ppm; 0.3 ppm; 1.2 ppm	[52]
	In vitro hepatocarcinoma cells (HepG2)	α-tomatine		1 μM	[92]
	In vitro hepatocarcinoma cells (HepG2)	α-tomatine		30 µM	[93]
	In vitro lung cancer cells (A549)	α-tomatine		1 µM	[94]
	In vitro breast cancer cells (MCF-7)	α-tomatine		7.07 µM	[95]
	In vitro non-small lung cancer cells (NCI-H460)	α-tomatine		2 µM	[96]
	In vitro prostate cancer cells (PC3)	α-tomatine		2 µM	[97]
	In vitro melanoma cells (BRAF, V600BRAF)	Tomatine (α-tomatine 87.1 ± 1.6%; dehydrotomatine 13.0 ± 0.8%)		1 µM	[98]
	In vitro leukemia cells (HL60, K562)	α-tomatine		5 µM	[99,100]
	In vivo Hasta strain rainbow trout	α-tomatine		100–2000 ppm	[101]

Abbreviations: DW: dry weight; FW: fresh weight; n.r.: not reported; *w*/*v*: weight/volume; p.o.: oral administration; i.v.: intravenous administration; s.c.: subcutaneous administration; i.p.: intraperitoneal administration.

## Data Availability

The data used to support the findings of this study are included in this article.

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
