# Peer review of "Agri-Food Waste Recycling for Healthy Remedies: Biomedical Potential of Nutraceuticals from Unripe Tomatoes (Solanum lycopersicum L.)"

_foods, 2024, doi:10.3390/foods13020331_

Round 1

Reviewer 1 Report

Comments and Suggestions for Authors

In this study, the authors reported a review concerning the nutritional properties of some compounds present in tomatoes, mainly SGAs. The manuscript is well-written, and it is easy to read. The bibliography reported is adequate

I have few suggestions for authors consideration:

Line 54. SGAs. Please, specify the acronym. The acronym is widely used throughout the document.

Lines 57 – 64. Please, provide references to support this information.

Similar consideration for lines 229 – 235.

Lines 323 – 324. Please, replace minerals by electrolytes. Minerals can only be considered Ca, Mg and P. It is a common mistake in the scientific literature that should be avoid.

Line 360. Please, provide the acronym of ROS

Line 370. “Several articles”. Please, provide the references of these manuscripts.

Line 423. Please, specify the type of cell line.

Author Response

Dear Reviewer 1,

Upon reviewers’ comments, we have improved the manuscript based on their suggestions and comments.

Comment: Line 54. SGAs. Please, specify the acronym. The acronym is widely used throughout the document.

Response: We thank reviewer for the correction. We added the meaning of the acronym at line 54.

Comment: Lines 57 – 64. Please, provide references to support this information.

Response: We thank reviewer for the correction. We provided two references to the sentences.

Comment: Similar consideration for lines 229 – 235.

Response: We thank reviewer for the correction. We provided a reference to the sentences.

Comment: Lines 323 – 324. Please, replace minerals by electrolytes. Minerals can only be considered Ca, Mg and P. It is a common mistake in the scientific literature that should be avoid.

Response: We thank reviewer for the correction. We replaced the word “minerals” with “electrolytes”.

Comment: Line 360. Please, provide the acronym of ROS

Response: We thank reviewer for the correction. We added the meaning of the acronym at line 362.

Comment: Line 370. “Several articles”. Please, provide the references of these manuscripts.

Response: We thank reviewer for the correction. The references of the papers which describe the anti-inflammatory effects of pure SGAs and green tomato extracts are reported in the subparagraph 5.2 (lines 372-396).

Comment: Line 423. Please, specify the type of cell line.

Response: We thank reviewer for the correction. We have added the cell culture models used for the evaluation of anti-tumoral activity in the line 425 (breast cancer (MCF-7), colon cancer (HT-29), gastric cancer (AGS), hepatocarcinoma (HepG2), and liver cancer (Chag)).

Reviewer 2 Report

Comments and Suggestions for Authors

The manuscript titled "Agri-food waste recycling for healthy remedies: biomedical potential of nutraceuticals from unripe tomatoes (Solanum lycopersicum L.)" is well-structured, with clear sections including an introduction, details on unripe green tomatoes as a source of glycoalkaloids, α-tomatine extraction methods, analysis methods, nutraceutical potential, and a conclusion. 

The language used is scientific, focusing on biochemical and pharmacological aspects. The manuscript follows a logical flow, starting from the introduction of the subject to more detailed discussions on specific compounds and their applications. It covers various topics, from the source of glycoalkaloids in unripe green tomatoes, α-tomatine extraction methods, and analytical techniques to the nutraceutical potential of green tomatoes. References are frequently used, which suggests a well-researched document.

However, there are a few areas where the authors might consider expanding or clarifying:

  1. While the manuscript focuses on unripe tomatoes, a comparative analysis with other fruit or vegetable sources of similar compounds could provide a broader context and enhance the manuscript's value.
  2. A more detailed section on the limitations of their study and potential areas for future research could be helpful. This might include the extraction process's scalability or the extracted compounds' stability.
  3. A discussion on the environmental impact of using agri-food waste for these purposes, including any sustainability considerations, could be a valuable addition.
  4. An economic analysis and environmental impact of the processes described, including cost-benefit aspects, could make the manuscript more relevant for industry professionals.

Line 19… A special attention was focused to highlight the [...] which should be revised for clarity and grammatical correctness​​.

Line 35 … As a consequence of this massive demand.. article

Line 83… The review summarised the literature data describing the extraction methods from waste the" is a bit difficult. A possible revision could be: "The review summarises literature data that describe the extraction methods from waste, the analysis protocols, and the main pharmacological activities of green tomato extracts [...]"​​.

Line 95.. SGAs are a class of secondary metabolites produced by some plants of Solanaceae family as tomatoes, potatoes, and eggplants." would be clearer as "SGAs are a class of secondary metabolites produced by some plants in the Solanaceae family, such as tomatoes, potatoes, and eggplants."​​.

Line 135.. The phrase "The protocols for unripe tomato analysis consist in adding the extraction mixtures to a determined amount of fresh tomatoes or lyophilized powered tomatoes and stirring the mixture for a certain period of time at controlled temperature." could be revised for clarity: "The protocols for unripe tomato analysis involve adding extraction mixtures to a specified amount of fresh tomatoes or lyophilized powdered tomatoes and stirring the mixture for a certain period at a controlled temperature."​​.

Line 325.. The phrase "Lycopene is endowed with antioxidant activity and in…" please rewrite.

Line 343.. There is a sentence fragment: "Extracts of leaves and immature green fruits of Californian Solanum lycopersicum L. show antimicrobial activity against Salmonella enterica, Staphylococcus aureus, and Escherichia coli K12." which could be restructured for better readability​​.

Comments on the Quality of English Language

The language used is scientific, focusing on biochemical and pharmacological aspects. 

Author Response

Dear Reviewer 2,

Upon reviewers’ comments, we have improved the manuscript based on their suggestions and comments.

Comment: There are a few areas where the authors might consider expanding or clarifying:

  1. While the manuscript focuses on unripe tomatoes, a comparative analysis with other fruit or vegetable sources of similar compounds could provide a broader context and enhance the manuscript's value.
  2. A more detailed section on the limitations of their study and potential areas for future research could be helpful. This might include the extraction process's scalability or the extracted compounds' stability.
  3. A discussion on the environmental impact of using agri-food waste for these purposes, including any sustainability considerations, could be a valuable addition.
  4. An economic analysis and environmental impact of the processes described, including cost-benefit aspects, could make the manuscript more relevant for industry professionals.

Response: We thank reviewer for the suggestions. Although these topics are interesting possible starting points for scientific research, to our best knowledge there are insufficient or no literature indications to be summarized in this review. Moreover, to clarify the scalability of the extraction process we have added the following sentences in lines 231-234: “Although α-tomatine extraction methods are reported in detail, to our best knowledge examples of scalability are not described in literature. Therefore, it is not possible to verify whether the previously described conditions represent an exhaustive system for glycoalkaloids extraction in a multigram scale.”

Comment: Line 19… A special attention was focused to highlight the [...] which should be revised for clarity and grammatical correctness​​.

Response: We thank reviewer for the correction. We have corrected the sentence at line 19 as “A special attention was given to the methods used”.

Comment: Line 35 … As a consequence of this massive demand.. article

Response: We thank reviewer for the correction. We have added the reference to the sentence.

Comment: Line 83… The review summarised the literature data describing the extraction methods from waste the" is a bit difficult. A possible revision could be: "The review summarises literature data that describe the extraction methods from waste, the analysis protocols, and the main pharmacological activities of green tomato extracts [...]"​​.

Response: We thank reviewer for the correction. We have clarified the sentence as follow: “The review summarises literature data which describe the extraction methods from waste, the analysis protocols, the main pharmacological activities of green tomato extracts titrated in α-tomatine”.

Comment: Line 95.. SGAs are a class of secondary metabolites produced by some plants of Solanaceae family as tomatoes, potatoes, and eggplants." would be clearer as "SGAs are a class of secondary metabolites produced by some plants in the Solanaceae family, such as tomatoes, potatoes, and eggplants."​​.

Response: We thank reviewer for the correction. We have corrected the sentence as follow: “SGAs are a class of secondary metabolites produced by some plants in the Solanaceae family, such as tomatoes, potatoes, and eggplants”.

Comment: Line 135.. The phrase "The protocols for unripe tomato analysis consist in adding the extraction mixtures to a determined amount of fresh tomatoes or lyophilized powered tomatoes and stirring the mixture for a certain period of time at controlled temperature." could be revised for clarity: "The protocols for unripe tomato analysis involve adding extraction mixtures to a specified amount of fresh tomatoes or lyophilized powdered tomatoes and stirring the mixture for a certain period at a controlled temperature."​​.

Response: We thank reviewer for the correction. We have corrected the sentence as follow: “The protocols for unripe tomato analysis involve adding extraction mixtures to a specified amount of fresh tomatoes or lyophilized powdered tomatoes and stirring the mixture for a certain period at a controlled temperature.”

Comment: Line 325.. The phrase "Lycopene is endowed with antioxidant activity and in…" please rewrite.

Response: We thank reviewer for the correction. We have corrected the sentence as follow: “Lycopene is characterized by an antioxidant activity and ameliorates cardiovascular symptoms, age related macular degeneration, and cataract in humans”

Comment: Line 343.. There is a sentence fragment: "Extracts of leaves and immature green fruits of Californian Solanum lycopersicum L. show antimicrobial activity against Salmonella enterica, Staphylococcus aureus, and Escherichia coli K12." which could be restructured for better readability​​.

Response: We thank reviewer for the correction. We have corrected the sentence as follow: “Leaves and immature green fruit extracts of Californian Solanum lycopersicum L. display antimicrobial activity against several bacteria (Salmonella enterica, Staphylococcus aureus and Escherichia coli K12)”.